# Clinical Decision-Making for Heart Failure in Kosovo: A Conjoint Analysis

**DOI:** 10.3390/ijerph192214638

**Published:** 2022-11-08

**Authors:** Ilir Hoxha, Besim Guda, Ali Hoti, Esra Zhubi, Erza Selmani, Blerta Avdiu, Jakob Cegllar, Dorjan Marušič, Aferdita Osmani

**Affiliations:** 1The Dartmouth Institute for Health Policy and Clinical Practice, Geisel School of Medicine at Dartmouth, Lebanon, NH 03766, USA; 2Evidence Synthesis Group, 10000 Prishtina, Kosovo; 3Research Unit, Heimerer College, 10000 Prishtina, Kosovo; 4Lux Development, 10000 Prishtina, Kosovo; 5General Hospital of Gjilan, 60000 Gjilan, Kosovo; 6General Hospital of Prizren, 20000 Prizren, Kosovo

**Keywords:** clinical factors, conjoint analysis, decision-making, evidence-based practice, healthcare reform, heart failure

## Abstract

Background: Heart failure represents a life-threatening progressive condition. Early diagnosis and adherence to clinical guidelines are associated with improved outcomes for patients with heart failure. However, adherence to clinical guidelines remains limited in Kosovo. Objective: To assess the clinical decision-making related to heart failure diagnosis by evaluating clinicians’ preferences for clinical attributes. Method: Conjoint analysis with 33 clinical scenarios with physicians employed in public hospitals in Kosovo. Setting: Two public hospitals in Kosovo that benefited from quality improvement intervention. Participants: 14 physicians (internists and cardiologists) in two hospitals in Kosovo. Outcome measures: The primary outcome was the overall effect of clinical attributes on the decision for heart failure diagnosis. Results: When considering clinical signs, the likelihood of a heart failure diagnosis increased for ages between 60 to 69 years old (RRR, 1.88; CI 95%, 1.05–3.34) and a stable heart rate (RRR, 1.93; CI 95%, 1.05–3.55) and decreased for the presence of edema (RRR, 0.23; CI 95%, 0.15–0.36), orthopnea (RRR, 0.31; CI 95%, 0.20–0.48), and unusual fatigue (RRR, 0.61; CI 95%, 0.39–0.94). When considering clinical examination findings, the likelihood for heart failure diagnosis decreased for high jugular venous pressure (RRR, 0.49; CI 95%, 0.32–0.76), pleural effusion (RRR, 0.35; CI 95%, 0.23–0.54), hearing third heart sound, (RRR, 0.50; CI 95%, 0.33–0.77), heart murmur (RRR, 0.57; CI 95%, 0.37–0.88), troponin levels (RRR, 0.59; CI 95%, 0.38–0.91), and NTproBNP levels (RRR, 0.36; CI 95%, 0.24–0.56). Conclusions: We often found odd and wide variations of clinical signs and examination results influencing the decision to diagnose a person with heart failure. It will be important to explore and understand these results better. The study findings are important for existing quality improvement support efforts and contribute to the standardization of clinical decision-making in the public hospitals in the country. This experience and this study can provide valuable impetus for further examination of these efforts and informing policy and development efforts in the standardization of care in the country.

## 1. Introduction

Heart failure is a complex clinical condition caused by any functional or structural heart defect that impairs ventricular filling or blood ejection to the systemic circulation to meet systemic needs [1,2]. Currently, it affects an estimated 23 million people worldwide [3]. Heart failure incidence increases with age, from approximately 20 per 1000 individuals 65 to 69 years old to >80 per 1000 individuals among those 85 years old. Although heart failure survival has improved over time, the absolute 5-year mortality rates from heart failure diagnosis have remained at 50% [4,5]. In Kosovo, heart failure is becoming increasingly prevalent. It manifests with frequent hospitalizations, poor quality of life, and a challenging diagnosis and treatment [6]. Arterial hypertension, often the cause of heart failure, is one of the major risk factors for mortality and morbidity [7,8]. Furthermore, hypertension is one of the independent predictors of poor exercise capacity for chronic stable heart failure patients, which often leads to high mortality rates [9,10]. Since 2014, hospitalization rates for hypertension have increased compared to the Organization for Economic Co-operation and Development (OECD) average of 33 nations in 2015 [11].

Heart failure is the chronic stage of any disease that leads to cardiac functional dysfunction, making it difficult to pinpoint a particular cause [12]. Multiple causes frequently coexist. Most comorbidities do not develop independently of heart failure. Instead, they share a set of risk factors that contribute to the syndrome’s pathophysiology or serve as a perpetuating reality [13,14]. The common risk factors for heart failure, such as coronary artery disease, hypertension, diabetes, obesity, and smoking, have been studied in several population-based cohorts [15]. Besides smoking, the burden of risk factors in patients with heart failure has been progressively increasing, with diabetes, obesity, and hypertension at the top of the list [13,16].

Heart failure remains a serious clinical and public health problem [17] and an economic burden as the overall number of patients living with heart failure rises, indicating the disease’s chronic path and population growth and aging [18]. Despite the advances in imaging technology and the increasing availability of diagnostic laboratory testing, patient history and physical examination remain the cornerstones in diagnosis and assessment of patients with heart failure [19,20]. Therefore, the current heart failure guidelines consider history, physical examination, and initial laboratory testing for biochemical markers as the guide for the early diagnosis and appropriate management of heart failure [21,22].

Using clinical guidelines and protocols is not common in the Kosovo healthcare system. Treatment of cardiovascular conditions is not an exception. There is a limited state-of-the-art treatment [10,11]. Clinical practice is based on the individual preferences of medical staff or historical practice patterns. There have been isolated efforts to develop and enact treatment protocols, including one for managing hypertension. Ministry of Health has established a unit for guiding and controlling the development of clinical guidelines and protocols and has worked with the support of donor-funded projects in this direction. The Project KSV/017 Health Support Program is one of them. It supports the quality improvement of health services in internal medicine at the two regional public hospitals of Prizren and Gjilan [23]. The quality improvement efforts focused on developing and using clinical standards of care, the standardization of processes within intervention hospitals, equipment, facility renovations, and continuous capacity building to implement clinical standards in care delivery. The project chose several clinical conditions (heart failure, among them) as the target of the quality improvement effort based on the number of patients with such conditions and interest from the hospitals for improving services for conditions of interest. The initial assessments noted practice variations in the decision-making for heart failure and other clinical decisions and procedures. To address this lack of standardization of clinical decision-making, the standard operating procedure (SOP) for diagnosis and treatment of patients with heart failure was developed, adapted, approved, and used by health care staff of both sites of intervention (public hospitals). In addition, training and other capacity-building for the use of SOPs for heart failure were provided to the medical staff. This intervention was expected to influence practice patterns in the decision-making for heart failure and the treatment of patients with heart failure.

To document practice variations and assess the early impact of the training (intervention), we have opted for conjoint analysis [24,25]. Conjoint analysis is a comprehensive method that facilitates hypothetical decision-making to allow examination of the effect of different attributes that would determine the respondent’s choice. It is a choice-based design with several hypothetical scenarios [24,25,26,27]. It enables examination of clinical decision-making at the micro-level (physician level) as opposed to the hospital (meso) or regional (macro) level. The micro-level focus helps us understand the direct causes of poor clinical decision-making and how it can reflect overall care patterns [28,29].

This study aims to understand the clinical decision-making related to heart failure diagnosis based on clinical signs or examinations by evaluating clinicians’ preferences through the discrete choice design of various hypothetical clinical scenarios. In addition, this study was interested in examining the quality measures’ early impact on reducing practice variations and improving service quality.

## 2. Methods

We performed two conjoint experiments. The idea of using conjoint analysis stems from the need to evaluate how a physician makes clinical decisions for a specific condition, such as heart failure, and if they consider the combination of potential attributes during the decision-making process. The first experiment examines the role of clinical signs (mainly from examination of patient history) on the decision for heart failure diagnosis. The second experiment examines the effect of clinical examination findings on the decision for heart failure diagnosis. Standard conjoint analysis methodology was used to design and implement this study, including five steps: defining attributes, assigning attribute levels, creating scenarios, obtaining preference data, and estimating model parameters. The conjoint analysis design is consistent with previous studies [24,25,27].

### 2.1. Definition of Attributes and Levels

A thorough literature review of textbooks and research papers through search databases such as Google Scholar, PubMed, and Scopus was performed to define each experiment’s attributes and attribute levels. This exercise was instrumental in identifying the most important and specific attributes related to heart failure. We chose attributes from symptoms, clinical signs, and biochemical markers of heart failure. The 2021 European Society of Cardiology (ESC) Guidelines for diagnosing and treating acute and chronic heart failure served as the main reference [30]. Interviews with internists and cardiologists were conducted to determine all the potential factors that can be used as attributes. After we collected feedback from internists and cardiologists, the number of attributes was reduced to an appropriate level. Each attribute was divided into two or more levels.

### 2.2. Design of Scenarios

The attributes were used to design the final scenarios, utilizing IBM SPSS Statistics V.22.0 software (IBM Corp., Armonk, NY, USA) orthogonal design facility. The orthogonal facility yielded a total of 32 scenarios for each experiment. An additional scenario (vignette) was added manually to each experiment for validation purposes, with features agreed upon in the literature as indications for heart failure.

### 2.3. Design of Questionnaire

We used structured interviews to collect the data. The questionnaire had two main sections. The first part of the questionnaire collected information about the participants (physicians) of the study, including socio-demographics, education, experience, and exposure to scientific work. The second and main part of the questionnaire contained all 33 scenarios of each experiment that physicians had to read and make hypothetical clinical decisions. Examples of vignettes are provided in the Appendix A. Prior to final use, the questionnaire was tested and reviewed accordingly.

### 2.4. Sample Selection and Data Collection

The sample consisted of internal medicine or cardiology specialist physicians working in both intervention hospitals. To be considered for the study, each participant was required to treat heart failure daily. A small sample size was deemed appropriate as it contained most respondents out of the total population. The data collection for each experiment was performed by an experienced researcher (I.H.) using in-person interviews in most cases. Online interviews were conducted when in-person interviews were not possible due to COVID-19 restrictions. After explaining the process for both experiments, during the first part of the interview, we collected data on the participant characteristics (age, gender, residence, experience, etc.) After that, in the central part of the interviews, the physicians were presented with a typical case history (so-called ‘paper patient’) and asked whether they would diagnose each scenario as heart failure. They had to examine and decide on each of the 33 scenarios of each experiment, in a total of 66 scenarios.

### 2.5. Data Analysis

We first performed a descriptive analysis of the key characteristics of respondents. For each experiment, the main analysis consisted of multinomial logistic regression models with robust variance estimates used to examine the effect of different clinical signs and attributes in the diagnosis of heart failure. Each clinical sign effect is represented by the Relative Risk Ratio (RRR). Statistical analyses were performed with STATA 17BE (Stata Corp., College Station, TX, USA). The study protocol has been reviewed by the Ethical Review Committee at Heimerer College with protocol number 2045/21. Although initially planned, we were restrained from performing sub-group analysis due to the small sample size.

## 3. Results

The final sample comprised 14 respondents (Table 1) out of 19 qualified staff members dealing with heart failure in daily practice in both hospitals. All participants were either practicing internists or cardiologists. To be certified in internal medicine, physicians in Kosovo should finish four years of residency training. Alternatively, they should finish five years of residency training in cardiology or sub-specialize in cardiology after becoming an internist, which would take six years of residency altogether. Hence, cardiologists have more considerable and specific experience managing cardiac conditions throughout their residency, presumably making them more qualified to diagnose and treat heart failure.

Only one of the participants was female (7%), and the rest were male (93%). Internal medicine was the specialty of 12 participants, while ten participants reported having a sub-specialty. Eight participants reported the use of approved SOPs. None of the participants regularly attended continuous medical education (CME) activities. More than half of the participants reported they had benefited from the training, but a smaller percentage reported they had also benefited from study visits. Regarding the ward’s project benefits, five of them reported their ward had benefited (36%). In contrast, nine respondents reported no project benefits of the ward in the form of equipment or renovation of working space (64%).

The attributes and levels (of attributes) used to generate case scenarios in both experiments are documented in Table 2 and Table 3. Several of these attributes demonstrated a statistically significant effect on the diagnosis of heart failure based on clinical signs or clinical examination scenarios (Table 4 and Table 5).

For the first experiment, consisting of signs of heart failure, participants decided that a case was a heart failure for 257 (56.6%) out of 462 cases presented to them. The age of 60–69 increased the likelihood of the diagnosis of heart failure compared to <60 and >70 old patients (RRR, 1.88; 95% CI 1.05–3.34). Furthermore, a stable heart rate also increased the chances of heart failure diagnosis compared to a decreased heart rate (RRR, 1.93; 95% CI, 1.05–3.55). On the other hand, the presence of edema decreases the likelihood of heart failure diagnosis compared to the non-presence of edema (RRR, 0.23; 95%CI, 0.15–0.36). Similarly, the presence of orthopnea and unusual fatigue decreases the chances of clinical diagnosis for heart failure compared to the lack of orthopnea (RRR, 0.31; 95%CI, 0.20–0.48) and unusual fatigue (RRR, 0.61; 95%CI, 0.39–0.94).

In the conjoint experiment with attributes consisting of clinical examination parameters, participants decided that a case was a heart failure for 276 (59.7%) out of 462 cases presented to them. The presence of high jugular venous pressure decreased the chance of heart failure diagnosis compared to the non-presence of elevated jugular venous pressure (RRR, 0.49; 95%CI, 0.32–0.76). Similarly, the presence of pleural effusion (RRR, 0.35; 95%CI; 0.23–0.54), the third heart sound presence (RRR, 0.50; 95%CI, 0.33–0.77), heart murmur presence (RRR, 0.57; 95%CI, 0.37–0.88) decreased the likelihood of the diagnosis for heart failure. The troponin levels at >0.4 ng/mL decrease the chance of diagnosis of heart failure compared to the troponin levels at 0–0.4 ng/mL (RRR, 0.59; 95%CI, 0.38–0.91). In addition, NTproBNP levels at <400 pg/mL also decreased the chance of the diagnosis of heart failure compared to NTproBNP levels at >400 pg/mL (RRR, 0.36; 95%CI, 0.24–0.56).

## 4. Discussion

Using the multi-attribute compositional models (conjoint analysis), we explored various factors associated with the clinical decision for heart failure diagnosis in Kosovo. The likelihood of heart failure diagnosis is influenced by age, heart rate, orthopnea, edema, and unusual fatigue. In addition, the heart failure diagnosis was less likely to happen in the presence of high jugular venous pressure, pleural effusion, third heart sound presence and heart murmur, higher troponin levels, and lower NTproBNP levels.

### 4.1. Context

Few studies evaluated the symptoms, clinical signs, shared decision-making, [30,31], and diagnostic procedures [32] to optimize patient care and heart failure outcomes. Moreover, a study by Pisa et al. [33] used conjoint analysis to assess patients’ treatment preferences. However, no existing study incorporated conjoint analysis in assessing the clinical decision-making among physicians regarding heart failure based on symptoms, clinical signs, and biochemical markers. 

### 4.2. Strengths and Limitations

The strength of this study is a design informed by a literature review and testing performed to identify attributes used in the analysis of the stated preferences (conjoint analysis). By incorporating the conjoint methodology to design and implement this study, we could generate attribute combinations and further evaluate how a physician would decide the diagnosis of heart failure. However, the conjoint analysis also has its limitations, including the limited experience of the respondents in understanding the process [24]. Furthermore, it assesses decision-making in an ideal context [23]. Nevertheless, previous studies report that this approach produces valid assessments of decision-making in clinical settings [23,24,25,26]. Another limitation is the modest sample size which, although representative of the population, did not allow for the generalization of results. As a result, although we planned to do a subgroup analysis, we hesitated to perform one and hence failed to understand if the project intervention had impacted service quality improvement in its early stages. For similar reasons, we hesitated to adjust the model with participant information. Data collection process was carried out during the critical times of the COVID-19 pandemic, which may have contributed to the fact that physicians have overlooked some typical attributes of heart failure.

### 4.3. Interpretation, Context, and Implications

It has been almost a quarter century since we entered the modern age of clinical guidelines to facilitate evidence-informed clinical decision-making, and several guidelines have been developed [34]. The diagnosis and treatment of heart failure are very well documented, with robust research and meta-analyses that support evidence-based practice [3,5,35]. Gender is a major determinant of heart failure diagnosis [36]. However not the focus of this study.

The results of this study show a considerable variation in clinical decision-making regarding the diagnosis of heart failure, reflecting a somewhat limited use of the guidelines, protocols, and scientific evidence among staff in public hospitals in Kosovo. This fact does not come as a surprise, considering that it is common in low-income and low-middle-income countries [37], where opportunities for professional advancement are limited. Similar results were found in another conjoint analysis performed in Kosovo investigating decision-making for appendectomy and pre-operative preparation for appendectomy [27].

The results also highlight the “skewness” of clinical decision-making for heart failure diagnosis toward factors, i.e., levels of attributes that are not correct or secondary for clinical diagnosis of heart failure (i.e., age, heart rate). In most studies, an increased high heart rate is associated with heart failure [38,39,40,41]. However, in conjunction with other factors, a stable heart rate does not exclude heart failure. We also see a preference for incorrect or secondary levels of attributes in orthopnea, edema, or unusual fatigue, which are known to be signs of heart failure. These findings can partially be explained by the participant’s level of understanding of the experiment, and partially by clinical decision-making preferences which are affected by attributes taken into consideration (in a case-by-case situation). In addition, we also observe a decrease in the relevance of clinical examination parameters that are commonly used in determining the diagnosis based on the most recent evidence-based practice (i.e., Troponin, NTproBNP). This detail reflects the lack of use of such parameters in daily practice and knowledge about them. Wide confidence intervals are also an important finding of this study, as they indicate the practice variation in clinical decision-making despite the small group of physicians included in the experiments.

Shifting from a traditional to a more evidence-based approach in clinical decision-making in a country like Kosovo is challenging and needs time. The potential for improvements is high with different incentives and activities in place. The adoption of SOPs can be regarded as an important starting point. However, the guidelines used in highly developed countries are not always applicable to the circumstances and situations in low-income countries (LIC) and low and middle-income countries (LMIC). Therefore, adjustments should be made and the inclusion of local experts and stakeholders is highly important and recommended [37]. Improvement of proper diagnosis and treatment according to clinical guidelines in public hospitals in Kosovo will have repercussions in improving existing residency programs. Such improvements can also shape continual medical education (CME) activities, the introduction of peer-to-peer review, and supervisory measures that ensure standardized clinical decision-making. 

This study can be a useful introduction to the problems with standardization of care using advanced designs like the conjoint analysis. This initial work can help generate awareness based on facts and trigger discussions for moving forward. Replicating this design to a wider group of physicians in other public hospitals could provide more meaningful and robust results. Replication of similar study design and analysis for other procedures can extend the examination of standardization of care in a wider spectrum of clinical procedures and hence provide deeper insights into the issues with standardization in the country. This whole expertise can be useful for countries with similar problems. 

## 5. Conclusions

Our study attempted to evaluate the clinical decision-making and use of standard operating procedures for heart failure. We often found odd and wide variations of clinical signs and examination results influencing the decision for heart failure diagnosis. These findings should be further investigated and understood as quality improvement efforts move forward. The study findings are important for existing quality improvement support efforts and contribute to the standardization of clinical decision-making in the public hospitals in the country. This experience and this study can provide valuable impetus for further examination of these efforts and informing policy and development efforts in the standardization of care in the country.

## Figures and Tables

**Table 1 ijerph-19-14638-t001:** Characteristics of the sample.

Characteristic	No. of Participants (N = 14)	%
Experience (in years)	23.50 *	8.26 **
Gender		
Female	1	7
Male	13	93
Urban residency		
Rural	1	7
Urban	13	93
Master of Science degree		
No	13	93
Yes	1	7
Type of specialty		
Internal Medicine	12	86
Cardiology	2	14
Hospital		
Hospital 1	9	64
Hospital 2	5	36
Working in the private sector		
No	6	43
Yes	8	57
Sub-specialty		
No	4	29
Yes	10	71
Regular attendance of CME activities		
No	14	100
Yes	0	0
Peer review involvement		
No	4	29
Yes	10	71
Five years of basic medical education ***		
No	6	43
Yes	8	57
Venue for medical studies		
Other	2	14
Prishtina	12	86
Length of residency program (in years)		
Four	4	29
Five	1	7
Six	9	64
Residency training venue		
Other	3	21
Prishtina	11	79
Benefited from training ***		
No	6	43
Yes	8	57
Benefited from study visits		
No	10	71
Yes	4	29
Using approved protocols		
No	5	36
Yes	8	57
Ward has benefited from the project		
No	9	64
Yes	5	36

Abbreviations: CME, continuous medical education. * Mean, ** Standard deviation, *** Respondents have finished medical school in a 5 or 6-year system. There was a shift in that system back and forth over the last 30 years. Benefiting means only the knowledge or skills gained during training or study visits attended by respondents during project implementation.

**Table 2 ijerph-19-14638-t002:** Attributes and levels for heart failure diagnosis—clinical signs.

Attributes and Levels
Age
<60
60–69
>70
Smoker status
Non-smoker
Former
Current
Other
Pain of discomfort in the chest
Palpitations
Visible jugular veins in the neck area
Family history of cardiovascular problems
No
Yes
Comorbidities (i.e., diabetes, hypertension)
No
Yes
Body Mass Index
Underweight
Normal
Overweight
Obese
Dyspnea
No
Yes
Reduced tolerance to physical effort
No
Yes
Persistent coughing or wheezing
Possible
Yes
Lack of appetite or nausea
No
Yes
Confusion
No
Yes
Edema
No
Yes
Heart rate
Decreased
Stable
Increased
Orthopnea
No
Yes
Unusual fatigue
No
Yes

**Table 3 ijerph-19-14638-t003:** Attributes and levels for heart failure diagnosis—clinical examination parameters.

Attribute and Levels
High jugular venous pressure
No
Yes
Hepato-jugular reflux
No
Yes
Pleural effusion
No
Yes
Displaced point of maximal impulse
No
Yes
Third heart sound can be heard
No
Yes
Heart murmur
No
Yes
Other symptoms
Ascites
Cachexia
Hepatomegaly
Irregular heart rhythm
Lung failure
Tachycardia
Tachypnea
C-reactive protein levels
1–3 mg/L
3–10 mg/L
>10 mg/L
Troponin levels
0–0.4 ng/mL
>0.4 ng/mL
Creatinine kinase MB levels
5–25 UI/L
>25 UI/L
Lactic acid dehydrogenase levels
140–280 UI/L
>280 UI/L
Brain natriuretic peptide levels
<100 pg/mL
>100 pg/mL
NTproBNP levels
<400 pg/mL
>400 pg/mL

**Table 4 ijerph-19-14638-t004:** Determinants of heart failure diagnosis—clinical signs.

Attribute	RRR	95% CI	*p*
Age			
<60	Ref		
60–69	1.88	1.05–3.34	0.03
>70	1.07	0. 64–1.78	0.77
Smoker status			
Non-smoker	Ref		
Former	0.85	0.45–1.61	0.63
Current	1.13	0.68–1.87	0.62
Other			
Pain of discomfort in the chest	Ref		
Palpitations	1.27	0.77–2.07	0.33
Visible jugular veins in the neck area	0.72	0.41–1.24	0.24
Family history of cardiovascular problems			
No	Ref		
Yes	0.89	0.58–1.39	0.63
Comorbidities (i.e., diabetes, hypertension)			
No	Ref		
Yes	0.77	0.50–1.19	0.24
Body Mass Index			
Underweight	Ref		
Normal	0.77	0.41–1.45	0.43
Overweight	0.74	0.39–1.38	0.34
Obese	0.67	0.37–1.22	0.19
Dyspnea			
No	Ref		
Yes	0.68	0.44–1.05	0.08
Reduced tolerance to physical effort			
No	Ref		
Yes	0.71	0.45–1.10	0.13
Persistent coughing or wheezing			
Possible	Ref		
Yes	0.71	0.46–1.11	0.13
Lack of appetite or nausea			
No	Ref		
Yes	0.99	0.64–1.54	0.99
Confusion			
No	Ref		
Yes	0.85	0.55–1.31	0.47
Edema			
No	Ref		
Yes	0.23	0.15–0.36	<0.001
Increased heart rate			
Decreased	Ref		
Stable	1.93	1.05–3.55	0.03
Increased	0.75	0.44–1.29	0.31
Orthopnea			
No	Ref		
Yes	0.31	0.20–0.48	<0.001
Unusual fatigue			
No	Ref		
Yes	0.61	0.39–0.94	0.02

Abbreviations: RRR, relative risk ratio.

**Table 5 ijerph-19-14638-t005:** Determinants of heart failure diagnosis—clinical examination parameters.

Attribute	RRR	95% CI	*p*
High jugular venous pressure			
No	Ref		
Yes	0.49	0.32–0.76	0.001
Hepato-jugular reflux			
No	Ref		
Yes	0.86	0.56–1.32	0.50
Pleural effusion			
No	Ref		
Yes	0.35	0.23–0.54	<0.001
Displaced point of maximal impulse			
No	Ref		
Yes	0.68	0.45–1.05	0.08
Third heart sound can be heard			
No	Ref		
Yes	0.50	0.33–0.77	0.002
Heart murmur			
No	Ref		
Yes	0.57	0.37–0.88	0.01
Other symptoms			
Ascites	Ref		
Cachexia	1.36	0.60–3.11	0.45
Hepatomegaly	1.34	0.64–2.77	0.42
Irregular heart rhythm	1.71	0.78–3.72	0.17
Lung failure	1.61	0.72–3.58	0.23
Tachycardia	1.06	0.44–2.53	0.88
Tachypnea	1.22	0.53–2.79	0.63
C-reactive protein levels			
1–3 mg/L	Ref		
3–10 mg/L	0.90	0.52–1.55	0.72
>10 mg/L	0.66	0.39–1.11	0.12
Troponin levels			
0–0.4 ng/mL	Ref		
>0.4 ng/mL	0.59	0.38–0.91	0.01
Creatinine kinase MB levels			
5–25 UI/L	Ref		
>25 UI/L	1.19	0.78–1.82	0.41
Lactic acid dehydrogenase levels			
140–280 UI/L	Ref		
>280 UI/L	0.98	0.64–1.50	0.93
Brain natriuretic peptide levels			
<100 pg/mL	Ref		
>100 pg/mL	0.79	0.52–1.22	0.30
NTproBNP levels			
<400 pg/mL	Ref		
>400 pg/mL	0.36	0.24–0.56	<0.001

Abbreviations: RRR, relative risk ratio.

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
