# Peer review of "Clinical Decision-Making for Heart Failure in Kosovo: A Conjoint Analysis"

_ijerph, 2022, doi:10.3390/ijerph192214638_

Round 1

Reviewer 1 Report

Dear author team,

Interesting publication you have submitted.

With much interest, I did read your manuscript and overall, I think your contribution is highly relevant, especially in a country context, where limited data availability is very common and where a general absenteeism of evidence-based practices in the public health sector can be widely observed. I cannot judge on the novelty/innovation of the study design and approach, as I am not an expert on that specific methodology. And I am not the best to judge if the suggested journal/special issue is the best place for publishing the article.

Overall I have some critical comments related to your manuscript, as I believe there is room for improvement to really make it a very fine contribution to the much needed data gaps in Kosovo.

Intro section:

1.  No mention on the history/current status of CGP in Kosovo. What are the efforts of the MoH?

2. No mention on the latest state of art of CGP, where of course you shall discuss the adaptation to the local Kosovo context. But reading the intro, the reader does not get the impression the author team is up to date with the latest discussions here, especially also in relation to gender as determinant for diagnosing heart failure…see also comment 7.

3. The intervention of LuxDev would benefit from a clearer description…SOPs on what? Training on what? Study visits on what? Quality improvement entailed what?

4. Why was heart failure chosen? What’s the epidemiological situation in Kosovo?

 Method section:

5. You mention an extensive literature review, but neither the procedure, nor the outcome of how you then further developed your experiment is well described. Also, the detailed description of the interviews is missing. The experiment development as such is presenting itself very week and needs revision.

6. About conjoint analyses technique, there are important differences amongst “market research” and “health” fields, as it relates to how it can be interpreted/used the feedback collected from the participants. Does this technique still validate the analyses coming from such a small sample?

7. Further, looking at clinical protocols for heart failure gender is a major determinant and it has not been looked at. There exist systematic reviews on that – already since 2011! See for e.g. Azad N, Kathiravelu A, Minoosepeher S, Hebert P, Fergusson D. Gender differences in the etiology of heart failure: A systematic review. J Geriatr Cardiol. 2011 Mar;8(1):15-23. doi: 10.3724/SP.J.1263.2011.00015. PMID: 22783280; PMCID: PMC3390064.

8. Could there be mentioned any relevant differences amongst cardiologists and internists, in relation to their specific residency programs, which could affect their role to diagnose/treat heart failure?

9. What are the comorbidities you looked at? These are major determinants and they are not mentioned.

10.  Explanation on your small sample size. Additional clarification on the number of case-histories (paper-patients) presented to the 14 research participants. How many of these cases were positive to the heart failure diagnose? Were all case-histories presented (and collecting responses) to all 14 participants?

Results section:

11.  All the tables would benefit from a leaner presentation. Why are you presenting %, with that small sample size for e.g.?

12.  It would be benefitting if tables number 3 and 4 could include the total subgroups by each category of attributes

13.  At the table 1, there is a variable “higher education” to which one participant has it, and the remaining 13 have not. If “higher education” does not correspond to high school level, what does that mean?

14.  At the same table 1, I suggest adding a variable that correspond to the professional life experience in the profession, either by age or number of years in profession.

15.  As per line number 161, the subgroup not benefitting from the project presents a RRR=0.53. If the other subgroup, which has benefitted from the project (line number 159) presents a RRR=3,42, then the overall group should have a RRR > 0.53 and not smaller (as per orthopnea row in the table 4).

Discussion:

16.  Generally, the correct anamnesis is the first important step in the treatment regime, and it would be even more interesting to assess if the specialist are taking the correct decision for treatment.

17.  Since you have experts from Slovenia on the author team, it would be interesting to get their input especially on the recommendation section, as that section would benefit from a more sincere outlook.

18.  Few of clinical and examination attributes used in the analyses are more than just risk factors, they are considered as diagnostic criteria for heart failure and missing them from the physicians should require a more careful analysis. The example could be the “orthopnea”. How is that possible for a cardiologist/internist to miss that sign in relation to heart failure? Is it due to the confusion with COVID-19, or the participants were not quite clear with their understanding of the process, which was already stated as one of the limitations of the study.

19.  The small sample could justify missing why many risk factors don’t support the likelihood of heart failure diagnose (corresponding to confidence interval including value of 1). But, finding out that several important risk factors were associated significantly to the alternative of missing this diagnose might raise critical doubt on the validity of the results. Probably the limitation, participants were not experienced enough to understand the process, could be more than just a limitation?!

Ethic review:

20.  To my understanding national ethic clearance has to be obtained, even in the case of an experimental study.

Reviewer 2 Report

The topic of this article is interesting. However, the grammar and spelling require extensive revision. There appear to be a lot of typos and ambiguities in the text. The authors should check it carefully.

Most importantly, I didn't see the point of using a conjoint analysis, nor a description of the process. Results were only descriptive or from logistic regression models. More information should be provided in the background section.

Line 24. Was there a typo of "odema"?

Line 122. Need to define SOP and CME when you mention it for the first time. There are many similar problems throughout the text.

Line 124. I suggest explaining what "project benefit" means in the article.

Table 1. When the authors refer to "5 years of basic medical education", does it mean more than 5 years or exactly 5 years? I suggest making this clear.

"Length of residency program" 4 years? 4 months?

"Benefited from trainings" "Benefited from study visits" It should be clear to what extent "benefit" can be defined. Did they learn more knowledge, earn a higher salary, or get better reimbursement?

Table 2. The words in the row "Increased heart rate" and belows were contradictory.

Table 4. What is RRR? Abbreviations should be defined.

Line 181. Has there been any studies in the past discussing the same issue? What are their conclusions? What are the advantages of your study compared to them? What have you done to overcome the limitations?

Lane 202. A high rate of what?

Line 215. The authors should further elaborate the confounding factors of the study.

Line 221. I suggest further explanation as to why the guidelines used in highly developed countries are not applicable to Kosovo and what adjustments can be made accordingly.

Will your research have any impact on society or the healthcare system in the future? Could your conclusions be applied to other diseases?

Reviewer 3 Report

Point 1: This study investigated the association between clinical attributes and heart failure diagnosis in a conjoint analysis with 33 clinical scenarios, which was executed with 14 physicians (internists and cardiologists) employed in two public hospitals in Kosovo. While the premise of this study is relevant and looks important, there are some comments/suggestions that authors may find helpful to improve the clarity and presentation of the manuscript.

Point 2: Abstract:- I would suggest adding a background statement summarizing the latest information on the topic and key phrases that pique interest.

Point 3: Abstract:- “The effect of clinical attributes 29 differed by project benefit (i.e. intervention) for both experiments”. Please elaborate a bit more on this point.

Point 4: There is little baseline knowledge on clinical decision-making for heart failure in the introduction. Authors need to develop further context and provide more information with reference to the guidelines for the diagnosis of heart failure and preceding analyses of practices in the region or worldwide and explain why their research is important.

Point 5: Fourteen respondents are unlikely a sufficient sample size, I am not sure if this study is powered enough for the study design and outcome measures.

Point 6: What type of interviews has been adopted in the present study (structured, semi-structured, or open-questioned? Please provide a synopsis of this point in the “Methods” section.

Point 7:
I would have liked it if the authors had merely offered a summary of the case scenarios  presented to participating physicians to decide whether or not they would diagnose heart failure as an appendix or supplementary document.

Point 8: Discussion - The authors should discuss in-depth, focusing on the interpretation of their results in the context of the previous evidence and the implication of the study findings.

Point 9: How can clinicians and health care authorities in Kosovo make use of the results presented herein to facilitate access to care and for policy-making? Consider commenting on this point in the “Discussion” section.

Point 10: Motivate on the study's merits, limitations, and how cautious readers should be while interpreting the results and give suggestions for future work.

Round 2

Reviewer 1 Report

Dear author team,

happy to read the improvement of your paper. 

For a final brush I am suggesting some language and style editing.

Author Response

Point 1. Dear author team, happy to read the improvement of your paper. For a final brush I am suggesting some language and style editing. Response 1. Thank you for your comments. We have found them very useful in the effort to improve our manuscript. Thanks for your suggestion. We have tried to perform a thorough final edit of the manuscript.

Reviewer 2 Report

Thanks for making the modifications. However, I still have some concerns as listed below.

Abstract. The objective of the study should be rewritten. It seems that the authors' ultimate goal was to assess physician adherence to clinical guidelines, and the investigation of the associations was only a means, not the primary purpose.

Line 57. What is OECD? Should define it when you first mention it.

Line 79. What does art treatment mean? 

Line 106. Micro is not an appropriate word here.

Line 159. What does IH mean?

Line 176. Should provide the approval number of the ethical review here.

Line 220. What does "change" mean here?

Line 297. Again, should define LIC and LMIC.

The authors mixed use of past tense and present tense in the manuscript, which should be corrected. Also, periods are missing from many sentences, making the paper hard to read.

Table 2 & Table 4. Should define what comorbidities include.

Author Response

Point 1. Abstract. The objective of the study should be rewritten. It seems that the authors ultimate goal was to assess physician adherence to clinical guidelines, and the investigation of the associations was only a means, not the primary purpose.

Response 1. Thank you for this comment. We agree that the primary intention was not the measurement of association. Hence we have tried to reflect better the aim of the study: To assess the clinical decision-making related to heart failure diagnosis by evaluating clinicians preferences for clinical attributes.

Point 2. Line 57. What is OECD? Should define it when you first mention it.
Response 2. Corrected. 

Point 3. Line 79. What does art treatment mean?
Response 3. Sorry for the lack of clarity here. State-of- the-art treatment is high-end care. The most advanced care available. It’s rather a common expression to outline that. So we prefer to keep it that way. If you or the editor don’t agree with that we can have it change to “high-end care” or “most advanced care available” during the later stages of manuscript processing.

Point 4. Line 106. Micro is not an appropriate word here.

Response 4. Thank you for your comment. But we disagree here. Micro, meso, macro classification is used by HSR Europe, to point out the interaction of different factors of care within the health care context. To make sure we provide enough explanation, we have specified that we mean physician level.
To make sure we reduce the chances of unclarity for the readers. we add references to the term we have used.
Europe, H. (2011). Health services research into European policy and practice.
Utrecht: Netherlands Institute for Health Services Research.
Europe, H. (2011). Health services research: helping tackle Europe’s health
care challenges. Policy Brief. Nivel: Utrecht.

Point 5.Line 159. What does IH mean?
Response 5. Ilir Hoxha. Abbreviation for one of the authors, which will be introduced at the beginning of the manuscript I assume after the paper if formatted. The abbreviation is already in the end.

Point 6. Line 176. Should provide the approval number of the ethical review here.

Response 6. Apologies for this omission, and thanks for bringing it to our attention. We have provided this information now: “…with the number of protocol 2045/21.”

Point 7. Line 220. What does 'change' mean here?

Response 7. It should have been chance. Thank you for noting that- it has been corrected.

Point 8. Line 297. Again, should define LIC and LMIC.

Response 8. Corrected.

Point 9. The authors mixed use of past tense and present tense in the manuscript, which should be corrected. Also, periods are missing from
many sentences, making the paper hard to read.

Response 9. Thank you for this comment. We have tried to perform a thorough review of all the manuscript to improve the language.

Point 10.Table 2 and Table 4. Should define what comorbidities include.

Response 10. Thank you for your comment. We have specified this now in both tables: Comorbidities (i.e. diabetes, hypertension)

Reviewer 3 Report

Comments of Manuscript ID: ijerph-1899221

After the criticisms and suggestions from the previous round were handled, the manuscript greatly improved.

I recognize the effort the authors put in to do this.

Minor comment

I noticed that the in-text citations are not sequentially numbered in the revised manuscript (particularly in the introduction section). The authors can be more careful to address this issue.

Author Response

Point 1. After the criticisms and suggestions from the previous round were handled, the manuscript greatly improved. I recognize the effort the authors
put in to do this.

Response 1. Thank you for your comments. We have found them very useful in the effort to improve our manuscript.

Point 2. I noticed that the in-text citations are not sequentially numbered in the
revised manuscript (particularly in the introduction section). The authors can be more careful to address this issue.

Response 2. Thank you for this comment. Sorry for our omission. We have tried to address this now.